# Dissecting the Brain/Islet Axis in Metabesity

**DOI:** 10.3390/genes10050350

**Published:** 2019-05-08

**Authors:** Esther Fuente-Martín, Jose M. Mellado-Gil, Nadia Cobo-Vuilleumier, Alejandro Martín-Montalvo, Silvana Y. Romero-Zerbo, Irene Diaz Contreras, Abdelkrim Hmadcha, Bernat Soria, Francisco Martin Bermudo, Jose C. Reyes, Francisco J. Bermúdez-Silva, Petra I. Lorenzo, Benoit R. Gauthier

**Affiliations:** 1Andalusian Center of Molecular Biology and Regenerative Medicine-CABIMER, Junta de Andalucia-University of Pablo de Olavide-University of Seville-CSIC, 41092 Seville, Spain; estherdlfuente@gmail.com (E.F.-M.); jose.mellado@cabimer.es (J.M.M.-G.); nadia.cobo@cabimer.es (N.C.-V.); alejandro.martinmontalvo@cabimer.es (A.M.-M.); irene.diaz@cabimer.es (I.D.C.); karim.hmadcha@cabimer.es (A.H.); bernat.soria@cabimer.es (B.S.); fmarber@upo.es (F.M.B.); jose.reyes@cabimer.es (J.C.R.); 2Instituto de Investigación Biomédica de Málaga-IBIMA, UGC Endocrinología y Nutrición. Hospital Regional Universitario de Málaga, 29009 Málaga, Spain; yaniromero@gmail.com (S.Y.R.-Z.); javier.bermudez@ibima.eu (F.J.B.-S.); 3Centro de Investigación Biomédica en Red de Diabetes y Enfermedades Metabólicas Asociadas (CIBERDEM), 28029 Madrid, Spain

**Keywords:** metabesity, T2DM, obesity, inflammation, pancreatic islet, astrocytes

## Abstract

The high prevalence of type 2 diabetes mellitus (T2DM), together with the fact that current treatments are only palliative and do not avoid major secondary complications, reveals the need for novel approaches to treat the cause of this disease. Efforts are currently underway to identify therapeutic targets implicated in either the regeneration or re-differentiation of a functional pancreatic islet β-cell mass to restore insulin levels and normoglycemia. However, T2DM is not only caused by failures in β-cells but also by dysfunctions in the central nervous system (CNS), especially in the hypothalamus and brainstem. Herein, we review the physiological contribution of hypothalamic neuronal and glial populations, particularly astrocytes, in the control of the systemic response that regulates blood glucose levels. The glucosensing capacity of hypothalamic astrocytes, together with their regulation by metabolic hormones, highlights the relevance of these cells in the control of glucose homeostasis. Moreover, the critical role of astrocytes in the response to inflammation, a process associated with obesity and T2DM, further emphasizes the importance of these cells as novel targets to stimulate the CNS in response to metabesity (over-nutrition-derived metabolic dysfunctions). We suggest that novel T2DM therapies should aim at stimulating the CNS astrocytic response, as well as recovering the functional pancreatic β-cell mass. Whether or not a common factor expressed in both cell types can be feasibly targeted is also discussed.

## 1. Introduction

Diabetes mellitus (DM) is currently a major social and economic burden worldwide. The global epidemic proportion of DM is one of the most important health problems of the 21st century, being the cause of four million deaths in 2017 [1]. According to the International Diabetes Federation, 425 million people were affected by DM in 2017 (8.8% worldwide prevalence), and this number is expected to rise to 629 million people by 2045 [1]. This alarming increase in the incidence of DM is mainly due to the increase in the number of type 2 DM (T2DM) sufferers, which accounts for 90–95% of all DM. T2DM is associated with environmental and nutritional factors, as well as lifestyle, operating on a genetic susceptibility background. Approximately 80% of T2DM patients are obese, and the increase in the global epidemic of obesity explains the dramatic explosion of T2DM incidence over the past two decades [1]. Over nutrition-derived metabolic dysfunctions, henceforth denoted as metabesity, contributes to the apparition of a chronic low-grade inflammation [2,3], which is recognized as the pathogenic link with T2DM. This inflammation initially targets peripheral tissues specialized in metabolism such as liver, adipose tissue, and pancreatic islets, and it plays a key role in insulin resistance and β-cell dysfunction [4,5,6]. Inflammation also affects the central nervous system (CNS), likely being one of the causes underlying the increased incidence of neurodegenerative diseases among T2DM patients. Of note, hypothalamic inflammation is arising as a key pathophysiological process in metabesity, linked to hypothalamic leptin and insulin resistance [7,8,9].

T2DM is a progressive metabolic disorder classically defined by chronic hyperglycemia due to the combination of an abnormal insulin secretion by pancreatic islet β-cells and increased insulin resistance of insulin-target tissues (adipose tissue, skeletal muscle, liver, and brain) [10]. Guidelines for T2DM management include, as first-line therapy, serious lifestyle interventions such as physical exercise, while long-term add-on therapies include medication in order to increase insulin secretion and sensitivity [11]. Although these treatments are adequate to improve hyperglycemia, they alleviate symptoms rather than target the root-cause of the disease and lead to the development of secondary complications [12]. For example, sulfonylureas that impel insulin secretion were shown to cause β-cell death and patients are thenceforth confined to daily insulin injections to control glucose homeostasis [13]. However, new classes of pharmacological agents such as sodium glucose transporter 2 SGLT2 inhibitors and glucagon-like peptide-1 (GLP-1) receptor agonists significantly improve the prognosis of T2DM patients [14]. Nonetheless, a better understanding of T2DM etiology is mandatory to develop more effective therapies to avoid long-term complications and early death associated with this disease, and to restrain the apparition of neurodegenerative disorders such as Alzheimer’s disease, dubbed T3DM [15]. In recent years, the CNS gained much interest as a key regulator of glucose/energy homeostasis [16]. Thus, T2DM is not only caused by failures in pancreatic β-cells but also dysfunctions in the CNS that could lead to the development of this disease. Moreover, metabesity-associated chronic inflammation also affects the CNS-mediated control of blood glucose levels, exacerbating the disease. Herein, we review the evidence that supports a main role for the CNS in glucose homeostasis, with a special focus on astrocytes, and how interventions that simultaneously target both the CNS and pancreatic islets may synergistically improve the regulation of blood glucose levels and metabolism in T2DM patients.

## 2. Brain and Pancreatic Islets in Glucose Homeostasis: Who Is in Control?

### 2.1. The Classical Pancreatic Endocrine Model

Glucose is a very important energy source for the body; thus, its circulating levels need to be within an adequate range to supply the metabolic demands and the proper function of the whole body. Failure in this glucose sensing/regulation process leads to metabolic disorders, including T2DM. The pancreatic β- and α-cells are key elements in the process of glucose sensing and homeostasis [17,18]. Pancreatic β-cells sense variations in plasma glucose levels and accordingly regulate both the synthesis and secretion of insulin through a process known as glucose metabolism/secretion coupling [19]. Pancreatic β-cells take up glucose via glucose transporter 2 (GLUT-2), which has a low affinity for glucose and high transport capacity, allowing a rapid equilibrium between extra- and intra-cellular glucose concentrations [20]. Glucose is then phosphorylated by glucokinase (GK), which acts as a glucose sensor and metabolic signal initiator of glucose-induced insulin secretion. Phosphorylation of glucose is the rate-limiting step in insulin secretion, and the importance of GK in glucose homeostasis is highlighted by the fact that mutations in the human glucokinase gene are linked to maturity onset diabetes of the young 2 (MODY2) [21]. Glucose phosphorylation is the first step of glycolysis leading to the production of acetyl CoA, which, in β-cells, is predominantly channeled into the Krebs cycle stimulating oxidative phosphorylation and ATP synthesis. Changes in the ATP/ADP ratio cause the closure of ATP-sensitive K^+^ channels, depolarization of the cell membrane, and opening of voltage-dependent Ca^2+^ channels [22]. Influx of calcium binds to synaptotagmin 7 (SYT7) located on insulin granules, triggering exocytosis [23]. Similar to β-cells, pancreatic α-cells also sense fluctuation in glucose, which in this case stimulates glucagon secretion. At low glucose concentrations, when the secretion of glucagon is stimulated, α-cells fire continuous overshooting action potentials [24,25]. Voltage-dependent Na^+^ channels and Ca^2+^ channels induce the upstroke of action potentials, opening voltage-gated Ca^2+^ channels, consequently elevating the intracellular Ca^2+^ concentration that triggers glucagon granule exocytosis and release [26,27]. Therefore, the exquisite regulation of both insulin and glucagon secretion and action is needed to maintain glucose levels within a physiological range and avoid detrimental effects.

When insulin levels are unable to maintain blood glucose levels within physiological ranges due to increased insulin resistance in peripheral tissues, glucose homeostasis is impeded, leading to T2DM. Despite the fact that T2DM is characterized by the development of insulin resistance, genome-wide association studies (GWAS) indicate that most T2DM susceptibility genes are involved in β-cell development and function [28]; among them, GK and GLUTs have a fundamental role in glucose sensing [20,29]. In addition, several factors indispensable for β-cell maturity/function, such as PDX-1, PAX4, and NEUROD, were identified [30,31,32,33,34,35,36,37]. Progression of T2DM characterized by a gluco-lipotoxic environment causes a decrease in the β-cell mass as a result of cell dedifferentiation and/or apoptosis, thus exacerbating the insulin deficit [38]. Accordingly, great efforts are focused on stimulating protection and/or regeneration of a functional β-cell mass to prevent the development of hyperglycemia. In this regard, we recently identified the chromatin factor HMG20A, a negative regulatory subunit of the LSD1/CoREST complex implicated in CNS development [39,40], as a master regulator of β-cell functional maturity. We found that HMG20A expression in islets is essential for metabolism/insulin secretion coupling via the coordinated regulation of key islet-enriched genes such as NEUROD, MAFA, and insulin, and that its depletion induces expression of genes such as PAX4 and REST implicated in β-cell de-differentiation as observed in T2DM islets [41]. Consistent with this, SNPs in the *HMG20A* gene are associated with T2DM and obesity in GWAS studies [42,43,44], and decreased expression of this factor was detected in islets from T2DM patients [41]. More recently, PAX8, for which SNPs are also associated with T2DM in GWAS, was identified as the first gestational diabetes mellitus (GDM) candidate gene critical for islet survival during pregnancy [45]. Additional genes such as *NR5A2* were shown to be important for rescuing β-cell death in islets isolated from donors with T2DM [46].

Notwithstanding the indispensable role of β-cells in the regulation of blood glucose levels, glucose homeostasis is maintained through the cooperation of different organs such as brain, pancreas, liver, skeletal muscle, gastrointestinal tract, and adipose tissue. These tissues nest specialized cells that continuously monitor blood glucose fluctuations [17,18,47,48] and, consequently, send hormonal and neuronal information to target tissues involved in glucose metabolism to normalize glucose levels. In this regard, a recent study showed that the CNS can play a key role in the coordination of the adaptive coupling of insulin secretion to insulin sensitivity in rodents [49], thus reinforcing the important role of the central glucose sensing in the maintenance of glucose homeostasis [48,50].

### 2.2. The Centralized, Brain-Based Model

The role of the brain in the control of glucose homeostasis and metabolism came into the limelight in recent years. Nevertheless, this is not a new concept, as, already in 1854, Bernard showed that puncturing the floor of the fourth brain ventricle (*piqure diabetique*) induced a fulgurate increase in blood glucose levels within 90 min of the intervention, causing a temporary artificial diabetes [51]. We now know that the hypothalamus is located in close proximity to the floor of the fourth ventricle and is the main cerebral site of convergence of several metabolic signals, involved in the regulation of feeding and fasting. Lesions in this area produce changes in body weight, food intake, and peripheral insulin levels [52,53,54]. The hypothalamus integrates peripheral hormonal and nutritional signals and orchestrates appropriate sympathoadrenal and neurohumoral responses to maintain glucose and energy homeostasis [17]. This brain region also coordinates hepatic insulin sensitivity and glucose production, as well as peripheral glucose uptake and insulin secretion [49,55,56]. Activation of brain fibroblast growth factor (FGF) receptors triggers a potent glucose-lowering response [57,58,59]. Under a situation of increased extracellular hypothalamic glucose, signals sent from the hypothalamus inhibit hepatic glucose production [60] and activate insulin secretion [61], resulting in the decrease of peripheral blood glucose levels. Alternatively, a hypoglycemic event can trigger a counter-regulatory response to restore normal glucose levels [62]. Enhancing further the important role of the hypothalamus in glucose homeostasis, the activation of hypothalamic inflammatory pathways correlates with the development of insulin resistance and T2DM [63,64]. This evidence reveals the important role for hypothalamic glucose sensing to maintain nutrient and glucose homeostasis, supporting the importance of a brain-centralized model in the regulation of blood glucose levels. Indeed, failure in the central sensing of carbohydrates leads to defects in the regulation of food intake [65], pancreatic β-cell function [66], and liver glucose homeostasis [67]. Therefore, a better understanding of this brain-centralized model and its cross-talk with the classical model of glucose homeostasis may highlight new opportunities for developing therapeutic strategies targeting metabesity.

## 3. Cell Types of the Hypothalamus Implicated in Energy Homeostasis

### 3.1. Hypothalamic Neurons

Several hypothalamic areas are involved in the control of energy homeostasis and glucose metabolism, including the arcuate (ARC), the ventromedial (VMH), the dorsomedial (DMH), the paraventricular (PVN) nuclei, and the lateral hypothalamic area (LHA) (Figure 1A) [68]. These nuclei possess glucose sensing neurons that adapt their electrical activity and excitability in response to alterations in extracellular glucose concentrations [48,69,70], triggering response signals to normalize the glycemia. These populations are divided into glucose-excited neurons (GE), which increase their electrical activity when extracellular glucose concentration increased, and glucose-inhibited neurons (GI) which are activated by a decrease in extracellular glucose levels (Figure 1B) [48,70,71]. GE neurons, located in the VMH, ARC, and PVN [71], use a similar glucose sensing mechanism as pancreatic β-cells [48,72], expressing GLUT-2 [73], GK [50], and K_ATP_ channels that, in response to glucose, lead to membrane depolarization and neurotransmitter exocytosis [74]. In addition, GE neurons can also use alternative glucose sensing mechanisms [73,75,76], such as the sodium-dependent glucose transporters SGLT-1 or-3, which are involved in glucose-induced neuronal activity in GE neurons [72,77,78]. Alternatively, the heterodimeric G-protein-coupled receptors of the T1R family, expressed in several hypothalamic areas [79], are also activated by glucose [72,77]. Moreover, recent studies suggest that GE neurons in the VMN nucleus respond to glucose load with mitochondrial responses that activate a signal transduction cascade that ultimately promotes lowering of systemic glucose [80]. The second neuronal population, GI neurons, which are mainly situated in the LHA, ARC, and PVN nuclei, increase their firing activity under hypoglycemic conditions [48,71]. The mechanism used by GI to sense glucose levels is less clear. One of the proposed models suggests that a reduction in glucose uptake decreases ATP production and Na^+^/K^+^ activity, resulting in the increase in cytoplasmic Na^+^, which leads to membrane depolarization via activation of a chloride conductance [81]. Alternative models include the involvement of other glucose transporters such as GLUT-1, GLUT-2, or GLUT-3 and AMP, whose concentrations are increased by low levels of glucose, leading to AMPK activation and neuronal firing [82,83]. Noteworthy, both GI and GE neurons co-exist in the same hypothalamic nuclei [84], and are also involved in the control of energy homeostasis and food intake. In this context, GI and GE hypothalamic neurons have a functional endocannabinoid system, which modulates their electrical activity in response to variation in energy and food intake [85,86].

The ARC nucleus, located in the floor of the third ventricle, plays an important role sensing peripheral signals that relay the systemic energy status [68]. Expression of receptors for metabolic hormones, such as leptin, ghrelin, and insulin, [68,87,88,89], and the capacity to sense several nutrients, including glucose and free fatty acids [69,71], allow this hypothalamic nucleus to respond to hormonal and nutritional inputs regulating energy homeostasis and metabolism to compensate for whole-body demands. A network of GE and GI antagonistic neuronal populations that together control the feeding behavior and energy homeostasis composes the ARC nucleus. GI neurons [90] express the orexigenic neuropeptide Y (NPY) and the agouti-related peptide (AgRP), which are stimulated under caloric restriction conditions [91,92] to induce feeding, inhibit energy expenditure, and regulate glucose metabolism [93,94,95]. In contrast, GE neurons [96] are activated in response to caloric excess to inhibit feeding and increase energy expenditure and weight loss [95,97]. This second population expresses anorexigenic proopiomelanocortin (POMC)-derived peptides such as α-melanocyte stimulating hormone (α-MSH) and cocaine-and-amphetamine-regulated transcript (CART). Animal model studies showed that alterations in these neuronal populations affect glucose metabolism in peripheral tissues [98,99]. Acute activation of AgRP neurons impairs both insulin sensitivity and glucose tolerance [100], and reduces energy expenditure [101], while the lack of AgRP neurons increases energy expenditure [102]. Additionally, intracerebroventricular (icv) administration of NPY, mainly produced by ARC neurons, induces insulin secretion [103]. POMC and AgRP neurons from the ARC project into the VMH, which senses hypoglycemia and induces a counter-regulatory response to restore normal glucose levels. This counter-regulatory response is mediated by the steroidogenic-factor 1 (SF1)-expressing neurons [62]. Ablation of these VMH SF1 neurons impairs the recovery from insulin-induced hypoglycemia, while their activation, under normoglycemic conditions, raises blood glucose levels by increasing glucagon secretion and inhibiting glucose-stimulated insulin secretion [55,62].

Although it is clear that GI and GE neurons play a key role in glucose sensing, these glucosensing neurons are not in direct contact with blood or cerebrospinal fluid, suggesting that other actors may be involved in this function [104,105]. In this regard, hypothalamic glial cells arise as central players for glucose sensing due to their strategic anatomical position connecting neurons with blood vessel and cerebrospinal fluid.

### 3.2. Hypothalamic Glial Cells

Virchow in 1846 initially coined the term neuroglia to describe a connective substance embedding neurons in the brain. Subsequently, the Spanish neuroscientist Ramón y Cajal reported on glial cells and their role in brain physiology [106]. The direct contact of glial cells to blood vessels and glucose sensing neurons raises the possibility that these cells mediate and bridge glucose signaling [105,107,108]. Indeed, the hypothalamic astrocytes and tanycytes, two types of glial cells, participate in the regulation of glucose and energy homeostasis [17,104,107,109] by modulating both peripheral and central glucose levels [60] and providing energy substrates to neighbouring neurons.

Tanycytes are specialised ependymoglial cells surrounding the lateral walls and floor of the third ventricle [110]. Cell bodies of tanycytes are in direct contact with the cerebral spinal fluid (CSF) and have a long process projected into the hypothalamus, a privileged position to sense hormones and nutritional signals from either the periphery or the CSF and relay them to the brain (Figure 1B) [110]. Astrocytes also have a strategic physiological location in close proximity to the blood–brain barrier (BBB) and, with their end-feet covering the surface of capillaries, support the hypothesis that astrocytes supply neuron energetic demands in a process known as the astrocyte–neuron lactate shuttle (Figure 1B) [107,111]. In the absence of glucose, the lactate derived from astrocytic glycogen metabolism maintains neuronal function [112].

Recent evidence highlights the important role of these two hypothalamic glial cells in glucose sensing and glucose homeostasis. Central icv injection of fibroblast growth factor 1 (FGF1) induces diabetes remission in animal models of T2DM in a process mediated by tanycyte activation that enhances insulin-independent glucose clearance [56,113]. Noteworthy, this effect of FGF1 administration is accompanied by a preservation of β-cell function [113]. Additionally, image analyses of rat brain slices showed that, in response to glucose, tanycytes display an increase in intracellular Ca^2+^ stimulating glycolysis and lactate release [110,114,115,116,117]. This lactate is then relayed to POMC neurons in the ARC triggering exocytosis of αMSH to produce satiety [118]. This mechanism is dependent on the GLUT-2, as inhibition of this transporter specifically in tanycytes disrupts the hypothalamic glucosensing mechanism, reducing glucose uptake and lactate production. Consequently, POMC neuronal activity is interrupted and the control of feeding behavior is altered [119]. Noteworthy, the astrocytic glucose transporters have also a critical role in glucosensing. As an example of this, the re-expression of Glut2 in astrocytes of Glut2 null mice restores glucagon secretion in response to hypoglycemia [120,121], indicating that hypothalamic astrocytes via GLUT-2 are a key part of the central glucose sensing machinery and play a key role in the regulation of glycemia. Likewise, repression of hypothalamic GLUT-1 levels due to sustained hyperglycemia blunted the decrease in systemic glucose production, whereas GLUT-1 over-expression in hypothalamic astrocytes restored glucose-sensing capacity of these cells [122]. Nevertheless, glucose transporters are not the only mechanism used by these glial cells to sense glucose. The fact that tanycytes also respond to non-metabolizable glucose analogs, 2-deoxyglucose and methyl glucopyranoside [115,117], suggests that additional glucosensing mechanisms are implicated. In this regard, G-protein-coupled receptors (GPCRs) were proposed to be implicated in glucose sensing and regulation [123]. Interestingly, the GPCR cannabinoid CB1 receptor, a well-known player in energy homeostasis, is expressed in astrocytes and facilitates neuron–astrocyte communication [124]. Although the role of the CB1 receptor in glucose sensing and homeostasis is still unknown, some evidence points to the regulation of the leptin signaling pathway and glycogen storage in astrocytes [125].

## 4. Astrocytes are Central to Glucose Metabolism and Homeostasis

Astrocytes, the most abundant cell type in the CNS, were initially described as passive supporters of neurons. However, recent data indicate that astrocytes are actively involved in CNS function. Astrocytes are essential for structural and nutritive support of neurons, synaptic transmission, neuronal plasticity and survival, regulation of neural immune responses, and neuroendocrine control of metabolism and energy balance [126,127,128,129,130,131,132]. The hypothalamic astrocytes form a network of cells connected via gap junctions that allow the rapid intercellular exchange of small molecules, including metabolic signals such as glucose and lactate [133,134]. Astroglial gap junctions are composed of connexin 30 and 43 (Cx30 and Cx43) for which expression levels are modulated by the metabolic status. Hyperglycemia increases Cx43 expression in the hypothalamus [135], leading to higher exchange of signals thereby modulating the neuronal glucose response. Small interfering RNA (siRNA)-mediated inhibition of hypothalamic Cx43 blunts pancreatic insulin secretion in response to a brain glucose challenge [135]. Moreover, astrogliosis, the morphological and functional changes of astrocytes in response to stress conditions, induces connexin changes [136]. Hence, obesity-related insults, such as hypothalamic inflammation, can potentially affect insulin secretion.

Astrocytes are the major metabolizers of glucose in the brain. These glial cells take up glucose through the insulin-independent glucose transporters GLUT-1 and GLUT-2. Glucose is either stored as glycogen or metabolized to lactate, which is then transferred by the monocarboxylate transporters (MTCs) to neurons as an energy substrate [107,137]. Lactate may also be the final metabolite sensed by neurons in response to increased glucose levels [60,138]. In line with this premise, brain lactate infusions decrease circulating glucose levels as a consequence of changes in the efferent neuronal signals controlling hepatic glucose production [60] and stimulate the hypothalamus–pancreatic axis responsible for the vagal control of insulin secretion [138]. These studies suggest that lactate release by astrocytes activates hypothalamic GE neurons in high-glucose conditions and modulate their responses in order to control glucose homeostasis, leading to a control of insulin secretion. Additionally the peroxisome-proliferator activated receptor *γ* (PPAR*γ*), which in astrocytes increases glucose uptake and lactate release to provide metabolic support for neurons [139,140], is also involved in glucose homeostasis. The specific lack of PPAR*γ* in murine astrocytes causes impaired glucose tolerance, correlating with increased hepatic expression of gluconeogenic genes such as pyruvate carboxylase, glucose-6-phospatase, or pyruvate dehydrogenase 4, and also several genes involved in lipogenesis and lipid transport and storage in liver [141]. Moreover, the disruption of astrocytic cholesterol synthesis alters brain development and function and results in systemic metabolic defects affecting carbohydrate and lipid oxidation at the whole-body level [142]. Therefore, lactate released by astrocytes has a key contribution to adequate insulin secretion and to the regulation of glucose homeostasis and lipid metabolism.

### 4.1. Hormonal Input Implicated in Astrocyte-Mediated Glucose Homeostasis

Nutritional changes, as well as the metabolic hormones insulin, leptin, and ghrelin, alter the sensitivity of the hypothalamic astrocytes [128,129,130,132,143,144,145,146], which express the respective receptors for these metabolic hormones [128,147,148] and modulate the neuronal circuits involved in metabolic control.

Ghrelin is a circulating hormone mainly produced and secreted by the stomach. Its acylated form stimulates the appetite and increases food intake by activating NPY and AgRP neurons in the ARC nucleus [149]. Ghrelin is also involved in both glucose sensing and homeostasis [150]. Although these effects are mainly mediated through ghrelin-responsive neurons in the hypothalamus, ghrelin has also a direct effect on hypothalamic astrocytes [128,132]. Treatment of astrocyte cultures with physiological concentrations of ghrelin increases the intracellular Ca^2+^ concentration [151] and modulates cytokine production [132]. Additionally, in vivo, chronic central administration of acylated ghrelin decreases astrocytic markers such as the glial fibrillary acidic protein (GFAP) or vimentin in rat hypothalamus [132]. The ghrelin receptor GHSR-1a is expressed in astrocytes of the ARC nucleus, and, through this receptor, acylated ghrelin modulates glucose uptake into hypothalamic astrocytes [128]. The icv administration of acylated ghrelin also modifies the expression of glucose transporters in the hypothalamus, which affects the glucose transport by astrocytes and could affect central glucose sensing [128]. We reported that ghrelin also stimulates the expression of glycogen phosphorylase, lactate dehydrogenase, and the monocarboxylate transport 4 in primary hypothalamic astrocytes [128]. The increase of these factors could lead to increased lactate production from glycogen stored in astrocytes, as well as to stimulate lactate transport to neurons. Altogether, these data indicate that the ghrelin effect on astrocytes modulates neuronal input signals, thus revealing new mechanisms via which astrocytes modulate glucose sensing and glucose homeostasis.

The adipocyte-derived hormone leptin regulates the metabolic status of the organism by affecting food intake and peripheral metabolic processes, including glucose utilization [99,152,153]. Accordingly, leptin-deficient mice have dysregulated glucose homeostasis that leads to hyperglycemia, hyperinsulinemia, and glucose intolerance [154]. In addition to the action of leptin on activation of POMC neurons and inhibition of AgRP neurons leading to satiety and increase in energy expenditure [155,156], this hormone has a direct effect on hypothalamic astrocytes that also express the leptin receptor. Leptin affects the morphology of hypothalamic astrocytes, as well as their capacity to capture glucose and glutamate [129,144]. Chronic icv administration of leptin in rats decreases the hypothalamic expression of GLUT-2, mainly expressed in astrocytes [129]. Although it is still unknown whether the effect of leptin on astrocytic glucose transport affects the neuronal glucose sensing, it is clear that leptin signalling in astrocytes modulates the metabolic response. Moreover, mice bearing an astrocyte-specific deletion of the leptin receptor exhibit a morphological decrease in glial coverage on the surface of hypothalamic POMC neurons and display an altered response to exogenous administration of leptin and ghrelin, as well as to fasting [157]. These data suggest that the contact between astrocyte and neurons is crucial for controlling neuronal synaptic inputs, which affect glucose sensing and systemic metabolism [157].

Insulin is a key hormone regulating glucose homeostasis. It acts both at the central and peripheral level. First evidence of its central actions came from Woods and Porte, who described that central injection of insulin in dogs induced a significant increase in pancreatic insulin output [158]. More recently, the role of insulin in regulating hypothalamic glucose sensing was reported [146]. Interestingly, astrocytes express the insulin receptor and relay neuronal glucose sensing. Loss of astrocytic insulin receptor diminishes glucose-induced activation of POMC neurons and alters glucose uptake and metabolism, leading to glucose intolerance [159,160]. Disruption in astroglial insulin signaling causes changes in the morphology of astrocytes, alters mitochondrial function, and reduces activation of POMC neurons, impairing the physiological response to glucose [160]. Thus, insulin signaling in astrocytes is crucial for central glucose sensing, as well as for systemic glucose metabolism via regulation of glucose uptake across the blood–brain barrier (BBB) [160,161,162]. Altogether, these data indicate that the metabolic hormones insulin, leptin, and ghrelin impact glucose uptake, transport, and metabolism via astrocytes [128,129,160].

### 4.2. Astrocytes and Neuroinflammation

Obesity-associated chronic inflammation affects the CNS, leading to neurodegenerative diseases and diabetes [63,163,164,165]. Indeed, hypothalamic inflammation is linked with impaired insulin release by pancreatic β-cells, as well as with insulin resistance in target tissues including the hypothalamus [166]. Important players in the hypothalamic inflammation associated with metabesity are astrocytes [167,168]. Central inflammation was linked with astrogliosis [167], a complex and multifaceted process characterized by an extensive proliferation of astrocytes with a reactive phenotype and an abnormal regulation of their functions. Although astrogliosis has an initial protective effect to preserve neuron functionality, a chronic long-term activation of astrocytes contributes to injury and to the development and progression of Alzheimer’s and DM [169,170]. In this sense, astrogliosis impairs insulin signaling and glycogen storage [171], increases glucose uptake and glycolysis [172], and alters glucose transport in the brain [173]. The morphological changes associated with astrocyte activation induce an increase in the glial coverage of POMC and NPY neurons, modifying neuronal synaptic inputs and affecting glucose sensing and systemic metabolism [157]. Moreover, astrogliosis can also promote neuronal dysfunction that likely leads to a variety of CNS disorders, including cognitive decline [174]. Taking this into account, central inflammation can induce degeneration of neuronal circuits beyond those impacting glucose and energy homeostasis to affect also those involved in cognitive process such as learning or memory. In agreement with this, epidemiological and clinical evidence suggests that metabesity and T2DM are associated with cognitive decrements, as well as increased risk for dementia [175,176]. In this sense, alterations in the insulin–central nervous system axis and brain glucose metabolism observed in T2DM patients are associated with neuronal death and with the progression of Alzheimer’s disease (AD) [160,173].

Neurodegenerative diseases linked to T2DM and metabesity are due to central inflammation, which involves changes in the production of cytokines and pro-inflammatory mediators [3,177]. One of these inflammatory mediators is the nuclear factor kappa B (NF-kB), which regulates the expression cytokines and chemokines through epigenetic and transcriptional mechanisms [178]. Interestingly, NF-kB is a target of the histone demethylase LSD1, a component of the LSD1/CoREST complex [179], and is stimulated during astrogliosis [7,169]. We recently demonstrated that HMG20A, a negative regulator of the LSD1/CoREST complex, is expressed in astrocytes. Furthermore, HMG20A expression was increased in brain of mice fed a high-fat diet correlating with astroglyosis that was characterized by increased GFAP, vimentin, and IL-1beta expression [180]. Transcriptome profiling of primary astrocytes in which HMG20A was silenced revealed a decrease in the inflammatory response, as well as in NF-kB-mediated inflammation [181]. Thus, similar to its role in promoting β-cell functional maturity (see Section 2.1), the expression of HMG20A in astrocytes may regulate the switch between non-reactive and reactive astroglyosis phenotype through potentiating the NF-κB/LSD1 complex activity.

## 5. Brain/Islet Glucose Homeostasis Axis Orchestrated by HMG20A?

Glucose homeostasis is maintained through a network of different organs and tissues that respond to alterations in glucose levels in an organized manner orchestrated by the brain. As described here, the hypothalamus senses glucose fluctuations and activates neuronal circuits that contribute to the regulation of insulin and glucagon secretion by pancreatic endocrine cells, as well as glucose metabolic pathways in liver, fat, and muscle, altogether resulting in the maintenance of glucose homeostasis. Nonetheless, a centralized model of glucosensing relies, as a last resort, on the adequate function of the pancreatic islets; thus, a brain–islet axis is indispensable for the fine-tuning of glucose homeostasis. Interestingly, both pancreatic β-cell and CNS cells possess similar developmental genetic programs that include common key transcription factors such as NEUROD and ISL1, as well as sharing identical regulated exocytosis machinery [23,182,183]. As such, potential epigenetic master regulators could foreseeably orchestrate both tissues/organs in cooperatively controlling glucose homeostasis. We and others showed that HMG20A is involved in neuronal and β-cell mature function [41,184]. As HMG20A regulates the expression of key genes such as NEUROD, GK, and GLUTs common to both astrocytes and β-cells, as well as genes such as insulin and GFAP restricted to each cell type, it is tempting to speculate that this chromatin remodeling factor is a common master regulator in β-cells and astrocytes, integrating inputs to altered glucose levels and stressful physiological conditions that can influence glucose metabolism (Figure 2). This premise is currently under investigation.

## 6. Conclusions

Accumulating evidence indicates that systemic glucose homeostasis is regulated, in part, through the CNS. Within this organ, hypothalamic astrocytes recently came into the limelight as the glucose sensor that facilitates the CNS response to changing metabolic environment. Nonetheless, current literature supports the notion that a cross-talk between brain and pancreatic islet is mandatory to maintain glucose homeostasis under various physiological or pathophysiological conditions, including metabesity and T2DM. Harmonizing the response of both cellular hubs, i.e., astrocytes and β-cells, to nutrient/glucose fluctuations will likely require a common denominator/sensor. We propose that such a sensor can be the chromatin-remodeling protein HMG20A, thus becoming an interesting target for reverting metabesity and diabetes (Figure 2). In this context, inhibitors of LSD1 that mimic the effect of HMG20A are currently in phase IIA clinical trials for the treatment of multiple sclerosis and Alzheimer’s disease (https://www.oryzon.com/en/news/2019). In parallel, we recently demonstrated that one such inhibitor, ORY-1001, could enhance expression of insulin, MAFA, NEUROD, and GK in the insulinoma INS-1E cell line [185,186]. These results highlight HMG20A, a new therapeutic target common to both the CNS/astrocytes and islet cells, which could be exploited to modulate the brain/islet axis in metabesity.

## Figures and Tables

**Figure 1 genes-10-00350-f001:**
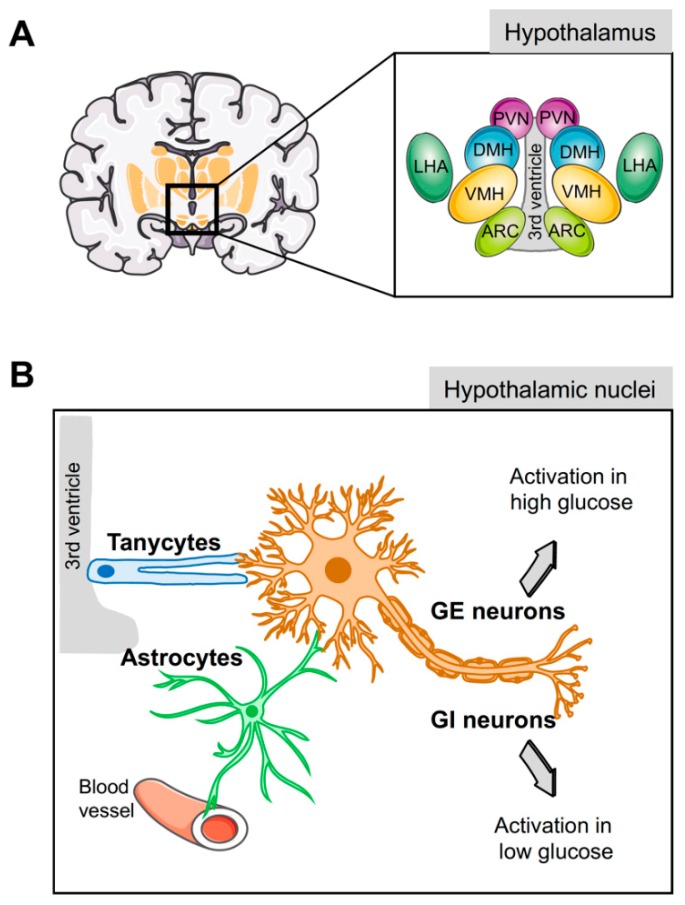
Glucose sensing by the hypothalamus. (**A**) Schematic representation of the different hypothalamic nuclei and their location surrounding the third ventricle. (**B**) Glucose-inhibited (GI) and glucose-excited (GE) neurons play a key role in glucose homeostasis; nevertheless, since these neurons are not located in direct contact with blood vessels or cerebrospinal fluid, sensing of circulating glucose levels is relayed through the hypothalamic glial cells (tanycytes and astrocytes). ARC, arcuate nucleus; VMH, ventromedial hypothalamic nucleus; DMH, dorsomedial hypothalamic nucleus; PVN, paraventricular nucleus; LHA, lateral hypothalamic area.

**Figure 2 genes-10-00350-f002:**
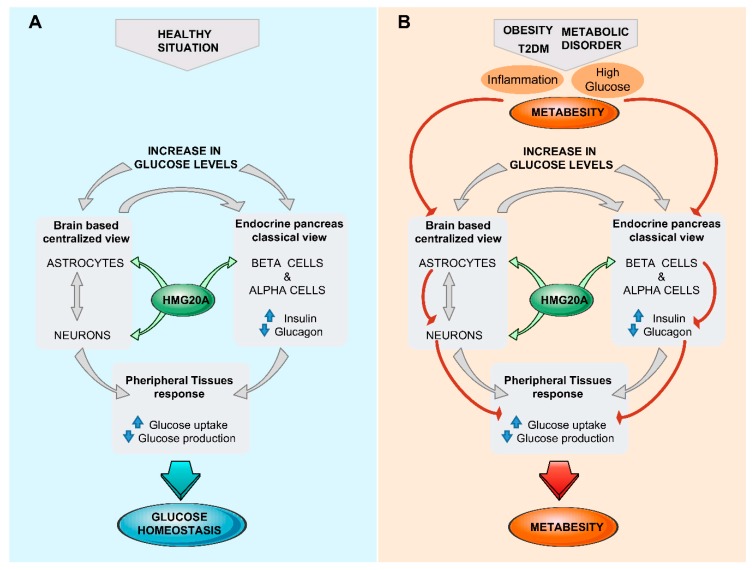
The brain (astrocyte)/islet axis in metabesity. (**A**) In response to increased glucose levels, a dialogue between brain astrocytes/neurons and islets coordinates glucose homeostasis by peripheral tissue relaying. The chromatin-remodeling factor HMG20A is a common denominator expressed in both astrocytes and islets that facilitates integration of input signals. (**B**) In the advent of metabesity, characterized by obesity, insulin resistance, and inflammation, which alter the expression of HMG20A, the brain/islet axis is short-circuited, precipitating the disease state. T2DM, type 2 diabetes mellitius.

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
