# Peer review of "Dissecting the Brain/Islet Axis in Metabesity"

_genes, 2019, doi:10.3390/genes10050350_

Round 1

Reviewer 1 Report

The goal of the review, as described in the abstract, was to discuss the chromatin remodeler HMG20A in its ability to „modulate the inflammatory response in astrocytes as well as to regulate the functionality of beta cells“... as „key target“ for „treatments for T2DM“.

This sounds a really interesting approach for a review.

The problem is, unfortunately, that the review is very immature at this stage, and sounds rather like an extract of a student’s thesis reused, as there is a long and unnecessary introduction on neurons and glial cells, partly quite unrelated to the topic, while HMG20A is only mentioned at the end, that it regulates the expression of key genes in b-cells and again, another single sentence on its effect in astrocytes.

HMG20A signals, targets, associated therapeutical strategies should be the center of the review and described in detail.

Chapter 3 reads really boring with all its morphological lists in 3.1 and thus is also difficult to understand-please add here a cartoon showing the brain with focus on the enlarged hypothalamus and its parts participating in glucose sensing. It reads kind of incoherently with no mechanistic input of the various neurons. This neuron part should be better structured, or shortened, as the intro says the aim is to focus on astrocytes?

Also the title of 3, must be improved: “Brain architecture in glucose sensing“-what does that mean? The chapter then talks only about Hypothalamic neurons and glial cells, while 4. continues on brain glucose sensing in astrocytes. Again, better structure is needed here.

Also a morphological cartoon for the glia cells in 3.2 would be helpful.

Check that chapter starting with line 379: “Insulin is considered the key hormone orchestrating glucose homeostasis in animals“

Sure. In animals. Also in humans? And anyway, this was said before above.

It acts both at the central and peripheral level. First evidences (evidence!) of their (many insulins? its!) central actions came from Woods and Porte, whom (?? who!) described that central injection of insulin in dogs induced a significant increase of pancreatic insulin output

Where are many more grammatical mistakes, including strange article use; please improve the following and probably other sentences:

Abstract;

Line 30: a process

Line 32: Approaches for? Novel

1.

Line 44: and this number

Line 46: as we are talking about human beings, avoid „cases“, which can be deleted anyway here

Line 56: inflammation arises as key...

Line 63: „newly diagnosed T2DM patients are prescribed an armada of pills that“

This is very floppy for a scientific article; there are very clear step-wise regulations for therapy (i.e. by the ADA, EASD), starting with serious life style interventions and such „armada“ may only be used at an end stage and surely not upon diagnosis. Needs to be improved!

Line 67: “for example sulfonylureas that force insulin secretion but also cause beta cell death: needs citation, e.g. JCEM 90(1):501–506, 2005

Line 77, 161 and other places: evidence; no plural

Line 78: with a special

Line 92: phosphorylated by glucokinase

Line 93: of glucose-induced

Line 109: due to a decrease in insulin secretion. Wrong! There is not a decrease in insulin, rather an increase. The problem here is non-compensated insulin secretion.

Line 116: Altogether these data point

Line 147: We now know that the hypothalamus is located in close proximity to the floor of the forth ventricle, which is...

I stop here, please review the rest of the article yourself with the help of a native speaker and/ or the senior author, who is usually excellent in paper writing.

Figure:

Typo: Disorder!

Metabesity and T2D should move to the end of the cartoon-downstream of the paths, as they are results of all the dysregulations, rather than causes???

Thereafter, a vicious cycle will be initiated and potentiates the problem.

Author Response

We thank the journal for its interest in our manuscript as well as both reviewers for their constructive comments. In this regards, the main comment/concern shared by both reviewers is related to the purpose of the review. We apologize for any confusion. The overall objective of the manuscript is to provide a comprehensive overview (for both experts and novice in either brain or islet-mediated glucose regulation) of the main cell types within the central nervous system as well as within the pancreatic islet that are implicated in glucose homeostasis, with a special emphasis on astrocytes and beta cells and how these two cell types coordinate glucose homeostasis. This general aim is now emphasized in the abstract. Consequently, a detailed description of the various neuronal subpopulations as well as glial cells involved in glucose/energy homeostasis is provided alongside astrocytes and beta cells. We then discuss and conclude the possibility that a common genetic denominator, such as HMG20A, may be implicated in the regulation of glucose and in the adaptive response to metabesity in both cell types. As such, we have replaced the last sentence of the abstract in order not to mislead the reader that the review is only focused on HMG20A. Nonetheless, as both reviewer expressed interest in HMG20A we have expanded its role and target genes in both islets and astrocytes without compromising our upcoming original manuscript reporting the role of HMG20A in astrocytes. As requested by reviewer 1, chapter 3 that describes the various CNS cell types has been revised and a new Figure 1 depicting a ‘cartoon showing the brain with focus on the enlarged hypothalamus and its parts participating in glucose sensing’ has been included to facilitate comprehension. In addition Figure 1 (now Figure 2) has been modified to reflect the interplay between the various cell types and HMG20A under physiological and pathophysiological situations. Finally the senior author has extensively revised/edited the manuscript including new section titles and references where appropriate. All changes are highlighted through the ‘Track changes’ function of Word. We also provide a clean copy to facilitate reading. We hope that the manuscript is now suitable for publication in Genes.

Point-by-point response to the reviewer’s queries

REVIEWER 1:

GENERAL COMMENT 1.1: The goal of the review, as described in the abstract, was to discuss the chromatin remodeler HMG20A in its ability to „modulate the inflammatory response in astrocytes as well as to regulate the functionality of beta cells“... as „key target“ for „treatments for T2DM“.

This sounds a really interesting approach for a review.

The problem is, unfortunately, that the review is very immature at this stage, and sounds rather like an extract of a student’s thesis reused, as there is a long and unnecessary introduction on neurons and glial cells, partly quite unrelated to the topic, while HMG20A is only mentioned at the end, that it regulates the expression of key genes in b-cells and again, another single sentence on its effect in astrocytes.

HMG20A signals, targets, associated therapeutical strategies should be the center of the review and described in detail.

RESPONSE 1.1: We apologize for the confusion. As stated above the overall objective is to provide a comprehensive overview (for both experts and novice in either brain or islet-mediated glucose regulation) of the main cell types within the central nervous system as well as within the pancreatic islet that are implicated in glucose homeostasis, with a special emphasis on astrocytes and beta cells and how these two cell types coordinate glucose homeostasis. This general aim is now emphasized in the abstract. Consequently, a detailed description of the various neuronal subpopulations as well as glial cells involved in glucose/energy homeostasis is provided alongside astrocytes and beta cells. We then discuss and conclude the possibility that a common genetic denominator, such as HMG20A, may be implicated in the regulation of glucose and in the adaptive response to metabesity in both cell types. As such, we have replaced the last sentence of the abstract in order not to mislead the reader that the review is only focused on HMG20A. Nonetheless, as both reviewer expressed interest in HMG20A we have expanded its role and target genes in both islets and astrocytes without compromising our upcoming original manuscript reporting the role of HMG20A in astrocytes.

COMMENT 1.2: Chapter 3 reads really boring with all its morphological lists in 3.1 and thus is also difficult to understand-please add here a cartoon showing the brain with focus on the enlarged hypothalamus and its parts participating in glucose sensing. It reads kind of incoherently with no mechanistic input of the various neurons. This neuron part should be better structured, or shortened, as the intro says the aim is to focus on astrocytes?

RESPONSE 1.2: As requested a new Figure ‘1’ depicting a cartoon showing the brain with focus on the enlarged hypothalamus and its parts participating in glucose sensing has been included to facilitate comprehension. Of note, we have acknowledged the use of the Servier Medical Art templates (http://www.servier.com) to generate figures.

Although the chapter seems boring and unstructured, we strongly believe that the morphological description of the various neuronal cell types implicated in energy homeostasis is important in order to have a global view of the various players without going into mechanistic input and guide the reader towards the importance of astrocytes in this complex cellular network. We sincerely attempted to structure this chapter as coherent as possible while keeping it short, which is not an easy task. In this context, we have shortened the paragraph on endocannabinoids to one sentence.

COMMENT 1.3: Also the title of 3, must be improved: “Brain architecture in glucose sensing“-what does that mean? The chapter then talks only about Hypothalamic neurons and glial cells, while 4. continues on brain glucose sensing in astrocytes. Again, better structure is needed here.

RESPONSE 1.3: We have amended the title to reflect the content of this chapter: Cell types of the hypothalamus implicated in energy homeostasis.

We then pursue with chapter 4 that focuses specifically on astrocytes.

COMMENT 1.4: Also a morphological cartoon for the glia cells in 3.2 would be helpful.

RESPONSE 1.4: A new Figure ‘1’ depicting a cartoon showing the brain with focus on the enlarged hypothalamus and its parts (various cell types) participating in glucose sensing has been included to facilitate comprehension.

EDITING COMMENTS:

Check that chapter starting with line 379: “Insulin is considered the key hormone orchestrating glucose homeostasis in animals“

The sentence was rephrased to: Insulin is a key hormone regulating glucose homeostasis.

Sure. In animals. Also in humans? And anyway, this was said before above.

‘Animals’ was deleted from the sentence.

It acts both at the central and peripheral level. First evidences (evidence!) of their (many insulins? its!) central actions came from Woods and Porte, whom (?? who!) described that central injection of insulin in dogs induced a significant increase of pancreatic insulin output

Thank you!! These mistakes were corrected accordingly.

 Their are many more grammatical mistakes, including strange article use; please improve the following and probably other sentences:

Abstract;

Line 30: a process, corrected

Line 32: Approaches for? Novel. Corrected

Line 44: and this number. Corrected

Line 46: as we are talking about human beings, avoid „cases“, which can be deleted anyway here. Corrected

Line 56: inflammation arises as key. Corrected

Line 63: „newly diagnosed T2DM patients are prescribed an armada of pills that“

This is very floppy for a scientific article; there are very clear step-wise regulations for therapy (i.e. by the ADA, EASD), starting with serious life style interventions and such „armada“ may only be used at an end stage and surely not upon diagnosis. Needs to be improved!

As both reviewers had concerns regarding this paragraph, we have rewritten it to include their suggestions along with appropriate references. Please refer to manuscript for changes.

Line 67: “for example sulfonylureas that force insulin secretion but also cause beta cell death: needs citation, e.g. JCEM 90(1):501–506, 2005.

Apologies as this reference was initially omitted and meant to be included.

Line 77, 161 and other places: evidence; no plural. Corrected

Line 78: with a special. Corrected

Line 92: phosphorylated by glucokinase. Corrected

Line 93: of glucose-induced. Corrected

Line 109: due to a decrease in insulin secretion. Wrong! There is not a decrease in insulin, rather an increase. The problem here is non-compensated insulin secretion.

We have amended the sentence to: When insulin levels are unable to normalize surges in blood glucose due to increased insulin resistance in peripheral tissues, glucose homeostasis is impeded leading to T2DM.

Line 116: Altogether these data point. Corrected

Line 147: We now know that the hypothalamus is located in close proximity to the floor of the forth ventricle, which is..

We have amended the sentence to: We now know that the hypothalamus is located in close proximity to the floor of the forth ventricle and is the main cerebral site of convergence of several metabolic signals, involved in the regulation of feeding and fasting.

I stop here, please review the rest of the article yourself with the help of a native speaker and/ or the senior author, who is usually excellent in paper writing.

The senior author has extensively revised/edited the manuscript.

Figure:

Typo: Disorder! Corrected

Metabesity and T2D should move to the end of the cartoon-downstream of the paths, as they are results of all the dysregulations, rather than causes???

Thereafter, a vicious cycle will be initiated and potentiates the problem.

We thank the review for his/her suggestion. We opted to divided the figure into two panels depicting (A) the Healthy situation and (B) the pathophysiological situation. 

Reviewer 2 Report

The authors are mainly reviewing roles of pancreas and astrocytes on glucose metabolism, and involvement of their focusing molecule, HMG20A. The theme is interesting and worth publishing. However, since the details of how pancreatic/astrocytic HMG20A affects on metabolism is not well described, readers are not able to understand the importance and significance of this molecule. By what experiments did the authors show what results, which suggests HMG20A's important roles in pancreas and brain?

Minor point: while described as "exogenous control of glycemia does not avoid the apparition of major secondary complications" in lines 69-70, some drugs such as metformin, SGLT2 inhibitors and GLP-1 receptor agonists were shown to significantly improve prognosis of diabetes patients. Therefore, this description seems an underestimation of medical treatments.

Author Response

We thank the journal for its interest in our manuscript as well as both reviewers for their constructive comments. In this regards, the main comment/concern shared by both reviewers is related to the purpose of the review. We apologize for any confusion. The overall objective of the manuscript is to provide a comprehensive overview (for both experts and novice in either brain or islet-mediated glucose regulation) of the main cell types within the central nervous system as well as within the pancreatic islet that are implicated in glucose homeostasis, with a special emphasis on astrocytes and beta cells and how these two cell types coordinate glucose homeostasis. This general aim is now emphasized in the abstract. Consequently, a detailed description of the various neuronal subpopulations as well as glial cells involved in glucose/energy homeostasis is provided alongside astrocytes and beta cells. We then discuss and conclude the possibility that a common genetic denominator, such as HMG20A, may be implicated in the regulation of glucose and in the adaptive response to metabesity in both cell types. As such, we have replaced the last sentence of the abstract in order not to mislead the reader that the review is only focused on HMG20A. Nonetheless, as both reviewer expressed interest in HMG20A we have expanded its role and target genes in both islets and astrocytes without compromising our upcoming original manuscript reporting the role of HMG20A in astrocytes. As requested by reviewer 1, chapter 3 that describes the various CNS cell types has been revised and a new Figure 1 depicting a ‘cartoon showing the brain with focus on the enlarged hypothalamus and its parts participating in glucose sensing’ has been included to facilitate comprehension. In addition Figure 1 (now Figure 2) has been modified to reflect the interplay between the various cell types and HMG20A under physiological and pathophysiological situations. Finally the senior author has extensively revised/edited the manuscript including new section titles and references where appropriate. All changes are highlighted through the ‘Track changes’ function of Word. We also provide a clean copy to facilitate reading. We hope that the manuscript is now suitable for publication in Genes.

Point-by-point response to the reviewer’s queries

REVIEWER 2:

COMMENT 2.1:

The authors are mainly reviewing roles of pancreas and astrocytes on glucose metabolism, and involvement of their focusing molecule, HMG20A. The theme is interesting and worth publishing. However, since the details of how pancreatic/astrocytic HMG20A affects on metabolism is not well described, readers are not able to understand the importance and significance of this molecule. By what experiments did the authors show what results, which suggests HMG20A's important roles in pancreas and brain?

RESPONSE 2.1: We apologize for any confusion. We have now expanded the experimental evidence demonstrating the key role of HMG20A as well as its downstream target genes in both islets (Mellado-Gil et al., Cell Death Dis 2018) and astrocytes (Fuente-Martin et al., manuscript in preparation). We then discuss the possibility that a common genetic denominator, such as HMG20A, may be implicated in the regulation of glucose and in the adaptive response to metabesity in both cell types.

COMMENT 2.2:

Minor point: while described as "exogenous control of glycemia does not avoid the apparition of major secondary complications" in lines 69-70, some drugs such as metformin, SGLT2 inhibitors and GLP-1 receptor agonists were shown to significantly improve prognosis of diabetes patients. Therefore, this description seems an underestimation of medical treatments.

RESPONSE 2.2:

As both reviewers had concerns regarding this paragraph, we have rewritten it to include their suggestions along with appropriate references. Please refer to the manuscript for changes.